# Long read single cell RNA sequencing reveals the isoform diversity of *Plasmodium vivax* transcripts

**Brittany Hazzard**[1], **Juliana M. Sá**[2], **Angela C. Ellis**[2], **Tales V. Pascini**[2], **Shuchi Amin**[2], **Thomas E. Wellems**[2], **David Serre**[1,3]*

1 Institute for Genome Sciences, University of Maryland School of Medicine, Baltimore, Maryland, United States of America, 2 Laboratory of Malaria and Vector Research, National Institute of Allergy and Infectious Diseases, National Institutes of Health, Bethesda, Maryland, United States of America, 3 Department of Microbiology and Immunology, University of Maryland School of Medicine, Baltimore, Maryland, United States of America

* dserre@som.umaryland.edu

**Data Availability Statement:** All sequence data generated in this study are deposited at the Sequence Read Archive under the BioProject PRJNA863611. https://www.ncbi.nlm.nih.gov/

## Abstract

*Plasmodium vivax* infections often consist of heterogenous populations of parasites at different developmental stages and with distinct transcriptional profiles, which complicates gene expression analyses. The advent of single cell RNA sequencing (scRNA-seq) enabled disentangling this complexity and has provided robust and stage-specific characterization of *Plasmodium* gene expression. However, scRNA-seq information is typically derived from the end of each mRNA molecule (usually the 3'-end) and therefore fails to capture the diversity in transcript isoforms documented in bulk RNA-seq data. Here, we describe the sequencing of scRNA-seq libraries using Pacific Biosciences (PacBio) chemistry to characterize full-length *Plasmodium vivax* transcripts from single cell parasites. Our results show that many *P. vivax* genes are transcribed into multiple isoforms, primarily through variations in untranslated region (UTR) length or splicing, and that the expression of many isoforms is developmentally regulated. Our findings demonstrate that long read sequencing can be used to characterize mRNA molecules at the single cell level and provides an additional resource to better understand the regulation of gene expression throughout the *Plasmodium* life cycle.

## Author summary

Single-cell RNA-sequencing is a valuable tool for identifying gene expression differences among cells present in one sample. However, scRNA-seq data is usually generated by sequencing the 3' end of mRNA molecules after poly-A capture, which complicates assigning reads to specific genes for organisms with poorly annotated UTRs, and prevents identifying differences in isoform expression. Here, we utilize a modified version of 10X scRNA-seq technology to characterize full-length transcripts using PacBio sequencing from both sporozoite and blood stages of *Plasmodium vivax*. These data allow us to predict full-length, stage-specific, transcripts for *P. vivax*, as well as to identify variations in UTR usage throughout the *P. vivax* life cycle.

bioproject/PRJNA863611/. Custom scripts are available through github at https://github.com/bhazzard11/Single-Cell-PacBio.

**Funding:** This work was supported by an award from the National Institutes of Health to the University of Maryland School of Medicine (U19 AI110820 to DS) and by the Intramural Research Program of the National Institute of Allergy and Infectious Diseases, National Institutes of Health. The funders had no role in study design, data collection and analysis, decision to publish, or preparation of the manuscript.

**Competing interests:** The authors have declared that no competing interests exist.

## Introduction

*Plasmodium vivax* is the second most common cause of human malaria worldwide and was responsible for 4.5 million clinical cases of malaria in 2020 [1]. Despite these numbers, *P. vivax* research lags behind that of *P. falciparum*, in part due to the difficulty to propagate the parasites *in vitro* [2,3]. Genomic techniques, such as genome [4–9] and transcriptome [10–12] sequencing, have improved our knowledge of *P. vivax* biology, but are hindered by the complexity of most blood-stage *P. vivax* infections: multiple genetically-distinct parasites are often simultaneously present in one infection and, due to the lack of, or incomplete, sequestration of *P. vivax* stages, all intraerythrocytic developmental stages concurrently circulate in the blood. Experimental infections of non-human primates using monkey-adapted strains of *P. vivax* [13,14] provide a robust system to study the regulation of all blood stages *in vivo* using monoclonal and well-characterized parasites. However, the simultaneous presence of multiple developmental stages in the blood, each with their own regulatory profiles, remains a major challenge. Short-term *ex vivo* cultures [10,11] and statistical inferences of the stage composition [10–12,15,16] have been used to circumvent this issue but suffer from limited resolution and possible artefacts.

Characterization of the gene expression of single cells (scRNA-seq) provides an elegant alternative and has been successfully applied to different *Plasmodium* species [17–23], including *P. vivax* [24,25]. However, many scRNA-seq assays rely on the capture and sequencing of the 3' ends of polyadenylated transcripts [26,27] and, consequently, only a short portion of each transcript is sequenced and the data generated provide little information about transcript isoforms and alternative splicing [11,12,15,28]. Additionally, it can be difficult to assign one signal to a specific gene since scRNA-seq reads typically derive from the 3' untranslated regions (3'-UTRs) which are incompletely annotated in *P. vivax* [11,12,15,24,28].

Full-length isoform sequencing (iso-Seq) using long-read technologies, such as PacBio, enables reliable characterization of gene isoforms and their UTRs [29–35]. This method has also been applied to single cells for identifying cell-type specific isoforms in a high throughput manner [30,34,35] and improving analysis of subsequent scRNA-seq data [34].

Here we combine long- and short read sequencing of scRNA-seq libraries to characterize *P. vivax* isoforms throughout the intraerythrocytic life cycle as well as in sporozoites. The standard short read Illumina sequencing of scRNA-seq libraries provides a robust description of the developmental stage of each *P. vivax* parasite captured, while sequencing the same mRNA molecules using PacBio long reads enables characterizing full-length transcripts present in these cells. The data generated allow to better annotate the 5'- and 3'-UTRs of *P. vivax* genes, to comprehensively characterize the transcripts expressed, and to identify stage-specific isoforms. Overall, our results demonstrate that single cell long read sequencing, in conjunction with short read sequencing, provides a robust method for comprehensively characterizing mRNA sequences at the single cell level and identifying isoforms involved in the regulation of specific *Plasmodium* stages.

## Results and discussion

### Characterization of single cell P. vivax blood-stage and sporozoite transcriptomes

We obtained blood-stage parasites from two *Saimiri boliviensis* monkeys infected with the Chesson strain of *P. vivax* and prepared 10X Genomics 3'-end single cell RNA sequencing (scRNA-seq) libraries after enrichment of infected red blood cells (see *Material and Methods*). After generating 57,550,235–63,399,045 short reads per sample, we successfully mapped 72–

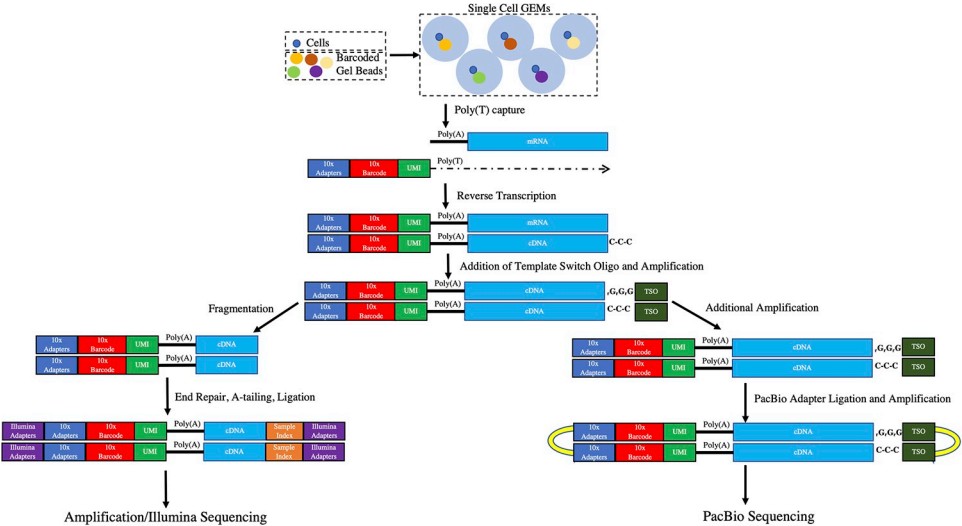

**Fig 1. Overview of the experimental design.** The figure shows, on the left, the standard 10X scRNA-seq protocol based on Illumina sequencing and, on the right, the protocol used for sequencing full-length transcripts using Pacific Biosciences chemistry. Abbreviations: GEMs, Gel Beads in Emulsion; UMI, Unique Molecule Identifier; TSO, template switch oligo.

74% of all reads to the *P. vivax* P01 genome sequence [36]. While full-length mRNAs are captured and converted into cDNA in the 10X droplets, only the 3'-end of each transcript is sequenced due to the cDNA fragmentation occurring during library preparation (**Fig 1**). All 3'-end scRNA-seq reads should therefore derive from the last ~300 bp of the transcripts (see e.g., **S1 Fig**). However, only 27–33% of the reads mapped within annotated *P. vivax* genes and 47–52% mapped more than 500 bp away from any annotated gene **S1 Table**). These results are consistent with previous analyses [24] and highlight that many genes and/or their 3'-UTRs remain incompletely annotated in the *P. vivax* genome. After removing PCR duplicates and stringent quality control filters, we obtained information about 949 and 1,807 single cell blood-stage transcriptomes, each characterized by more than 5,000 unique reads (**Table 1**). To characterize which blood stages were present in each infection, we analyzed these transcriptomes using principal component analysis and showed that several asexual and sexual developmental stages were present in both samples, although with some differences in their relative proportions (**S2 Fig**).

We also prepared two scRNA-seq libraries from salivary gland sporozoites dissected from *Anopheles stephensi* and *Anopheles freeborni*, respectively, fed on *Saimiri* monkeys infected

**Table 1. Illumina and PacBio sequencing results and transcript predictions.**

| | Illumina sequencing data | | | | PacBio sequencing data | | | |
|---|---|---|---|---|---|---|---|---|
| Sample ID | No. of reads | %Pv reads | No. of unique reads^ | No. of cells* | No. of reads (CCS) | %in Illumina cells# | No. of unique reads^ | No. of transcripts |
| Blood1 | 57,550,235 | 74% | 31,177,026 | 949 | 3,390,270 | 64% | 1,688,745 | 9,982 |
| Blood2 | 63,399,045 | 72% | 35,706,126 | 1,807 | 3,570,838 | 50% | 1,523,456 | |
| Spz1 | 63,039,836 | 7% | 3,654,760 | 2,609 | 2,578,778 | 8% | 106,531 | 784 |
| Spz2 | 74,606,583 | 3% | 2,131,303 | 2,363 | 1,935,640 | 5% | 40,977 | |

^No. of mapped reads with unique molecular identifiers (UMIs).

#Percentage of PacBio reads with a GEM barcode corresponding to one of the Illumina-defined single cell transcriptomes.

with *P. vivax* parasites. Out of the 63,039,836 and 79,688,059 reads generated from each library, 3–7% mapped to the *P. vivax* genome sequence, similar to the numbers obtained previously for *P. berghei* sporozoites [21]. 25–26% of those reads mapped within annotated *P. vivax* genes and 58–59% mapped more than 500 bp away from any gene (**S1 Table**). After removing PCR duplicates and stringent quality control filters, we obtained information about 2,609 and 2,363 single cell transcriptomes, each characterized by more than 250 unique reads (**Tables 1 and S1**). We used a lower threshold for the sporozoite analysis due to the lower number of *P. vivax* reads recovered as a result of the overwhelming presence of mosquito RNA (but the cutoff is comparable to those previously used for analyzing sporozoite scRNA-seq data [21]). In contrast to the heterogeneity of the blood stage parasites, these salivary gland sporozoites formed a relatively homogeneous population (**S2 Fig**).

## Long read sequencing of scRNA-seq libraries provides full-length transcript information

Since only the 3'-end of each transcript is usually sequenced in scRNA-seq experiments, it is sometimes difficult to assign the scRNA-seq reads to a specific gene, or to identify signals derived from different gene isoforms [24]. We therefore used cDNA from the same four 10X scRNA-seq libraries before fragmentation to generate full-length isoform sequences using the PacBio chemistry (see **Fig 1** and *Material and Methods* for details).

From each blood sample, we generated 3,390,270 and 3,570,838 circular consensus sequences (CCS) that each derived from at least 10 passes of sequencing of an individual cDNA molecule. 50–64% of these sequences carried a 10X barcode matching one of the cells characterized by Illumina sequencing, and >99% of those sequences mapped unambiguously to the *P. vivax* genome (**Tables 1 and S1**). After removal of PCR duplicates, we ended up with 1,688,745 and 1,523,456 unique reads (each derived from a unique mRNA molecule) for further analysis (**Table 1**). While the Illumina and PacBio library preparations diverged after cDNA amplification, the same barcoded cDNA molecules were used for both experiments (**Fig 1**) and the numbers of reads obtained with each technology for each individual cell were highly correlated (**S3 Fig**).

For the sporozoite samples, we generated 2,578,778 and 1,935,640 circular consensus sequences. 5–8% of these sequences carried a 10X barcode matching one of the cells characterized by Illumina sequencing, and 50–65% of those sequences mapped unambiguously to the *P. vivax* genome. After removal of PCR duplicates and additional QCs, we ended up with 106,531 and 40,977 unique reads respectively for further analysis (**Tables 1 and S1**). Similar to the pattern observed for the blood stage samples, the numbers of reads obtained by Illumina and PacBio sequencing for each sporozoite were highly correlated (**S3 Fig**) although, due to the lower sequencing depth, the overall number of cells with PacBio information was limited to 2,449 and 1,670.

We then summarized these mapped reads into *P. vivax* transcripts and, to avoid including sequences that may represent partially degraded molecules or technical artefacts, we only considered predicted transcripts represented by more than 10 PacBio unique reads in both samples of the same type (e.g., in both blood-stage samples). Overall, we identified a total of 9,982 transcripts from blood stage parasites and 784 from sporozoites (**Tables 1 and S2**), with an average transcript length of 2,432 bp from blood stages and 2,054 bp from sporozoites and ranging from 100 to 20,506 bp and from 197 to 6,809 bp, respectively. (Note that the transcript prediction algorithm occasionally leads to artefacts when transcripts in the same orientation overlap, which seems to be the case for the 20 kb transcript.)

## Most transcripts encode proteins corresponding to the genome annotation

Out of the 9,982 blood stage transcripts, 8,550 transcripts were predicted to encode more than 100 amino acids and 5,368 of those (63%) were predicted to encode a full-length protein (including a start and stop codon) (**S1 Table**). The median length of these full-length protein sequences was 291 amino acids, significantly smaller than the length of the protein sequences annotated in the latest version of the P01 *P. vivax* genome [36] (median of 401 amino acids) (**S4 Fig**). This discrepancy is possibly due to the lower 10X reverse transcriptase processivity that may hamper synthesis of extended cDNA molecules and prevent recovery of transcripts from long genes (note that most full-length transcripts obtained from PacBio sequencing match the annotated reference, see below). Of the 784 sporozoite transcripts, 563 transcripts were predicted to encode more than 100 amino acids and 344 of those (60%) were predicted to encode a full-length protein (including a start and stop codon), with a median length of 267 amino acids (**S4 Fig** and **S1 Table**).

We then compared the protein sequences predicted from these full-length transcripts to all *P. vivax* protein sequences annotated in the P01 genome (see **S5 Fig,** *Materials and Methods*). Most transcripts were predicted to encode protein-coding sequences highly similar to those annotated in the genome: 4,553 of the blood-stage transcripts (84.8%) and 254 of the sporozo-ite transcripts (73.8%) were identical to an annotated *P. vivax* protein-coding sequence for >90% of their length (**S6 Fig,** see e.g., **Fig 2A**). Additionally, 378 blood-stage (7.1%) and 23 sporozoite transcripts (6.7%) partially matched known protein-coding sequences and were identical over 50–90% of their sequence. While some of these transcripts could represent instances where the current gene annotation is possibly incorrect (see e.g., **Fig 2B**), in 253 of these cases (63%) another isoform in our dataset matched the entire annotated protein-coding sequence, suggesting that these differences represent incomplete annotations of transcript iso-forms rather than incorrect annotations (see also below). One example is the cytochrome b5-like heme/steroid binding protein (PVP01_0716500) that is transcribed as annotated in the genome by blood-stage parasites but, due to an alternative start site, is transcribed into a shorter mRNA by sporozoites, resulting in a shorter predicted protein (**Figs 2C and S7**). Finally, 436 blood-stage (8.1%) and 67 sporozoite transcripts (19.5%) aligned over less than 50% of their length to annotated protein sequences but those transcripts were very short (less than 200 amino acids) and/or had low support from the PacBio reads and likely represented artifacts or fragmented transcripts. The complete list of predicted transcripts, their sequence and their identity to reference transcripts is presented in **S2 Table**.

## Most P. vivax transcripts have extended UTRs that often contain introns

Even for the transcripts highly similar to the annotated protein-coding genes, the PacBio sequences add novel information by providing a detailed description of the 5'- and 3'-UTRs which, despite recent efforts [28], remain incompletely annotated in *P. vivax*. Consistent with previous reports [11,15], our data showed extensive UTRs in many genes (**Fig 3**). We observed that 5'- and 3'-UTRs were roughly similar in length in transcripts expressed by blood-stage parasites, while 5'-UTRs were, on average, slightly longer than 3'-UTRs in sporozoite tran-scripts (745 vs 731 bp in blood stages [p-value = 0.3149] and 762 vs 650 bp in sporozoites [p-value = 0.0076]). To evaluate how this improved characterization of UTRs would affect future analyses of scRNA-seq data, we compared the percentage of scRNA-seq reads mapped to the currently annotated *P. vivax* genes with the percentage mapped to annotations supplemented by our predicted isoforms (available in **S1 Data**). While only 27–33% of the scRNA-seq reads mapped within previously annotated genes for the blood stage parasites and 25–26% for the sporozoites, including the PacBio predictions raises these figures to 69–77% and 65–70%,

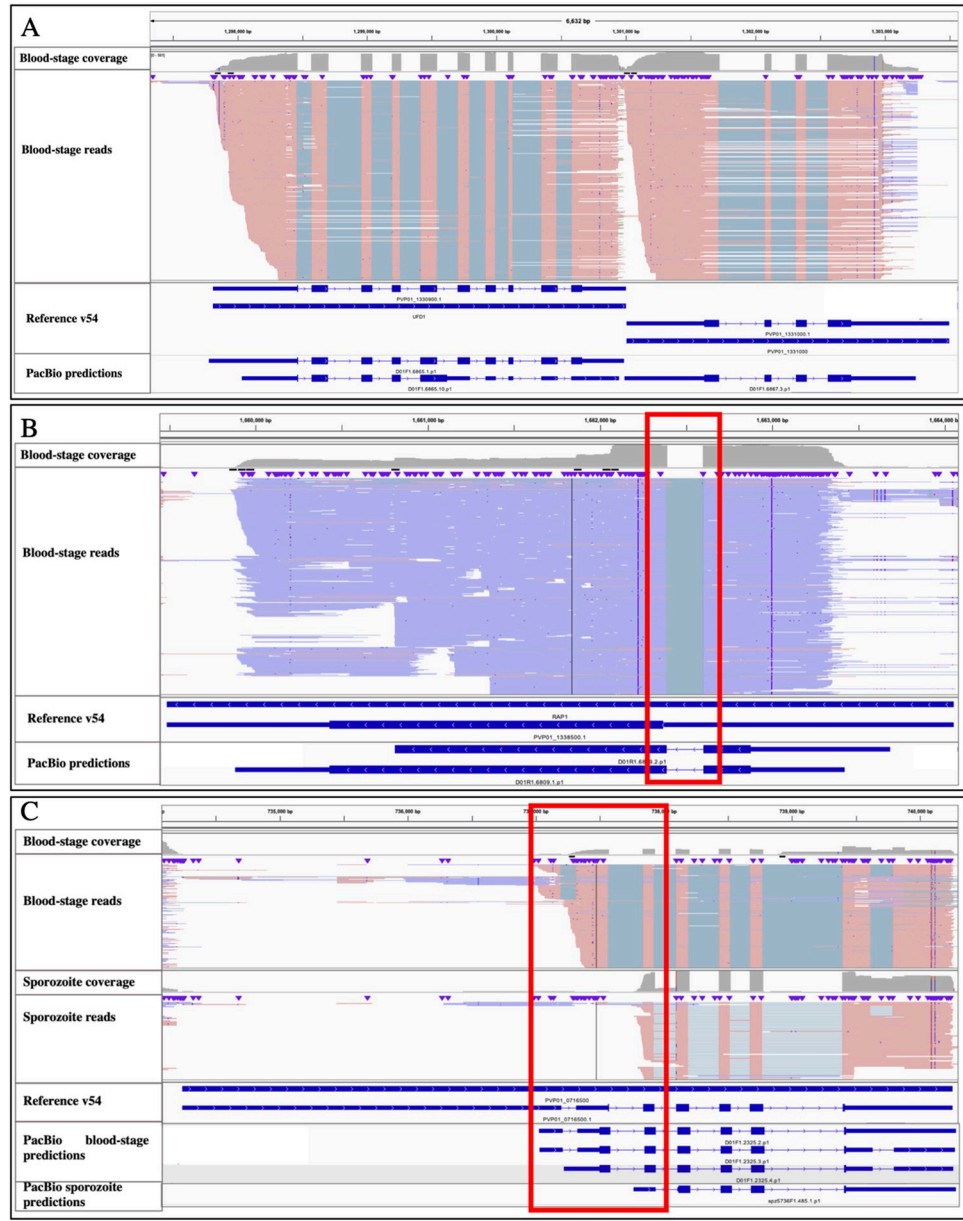

**Fig 2. Examples of full-length isoform sequencing using PacBio and resulting transcript predictions.** (**A**) Example of protein-coding transcripts matching the current gene annotations. The figure shows 5.7 kb of chromosome 13 containing two annotated *P. vivax* genes, the ubiquitin fusion degradation protein 1 (PVP01_1330900) and an ATP synthase-associated protein (PVP01_1331000). The blue horizontal bars at the bottom shows the annotations for these genes in plasmoDB v54, while the top panel shows the PacBio reads mapping to this locus (each red horizontal line is a unique read mapped to the positive strand with the grey lines indicating spliced introns). Note that, while the PacBio reads support a shorter 3'-UTR than annotated for PVP01_133100, the predicted protein coding sequences are identical to the ones annotated. (**B**) Example of protein coding transcript differing from the current gene annotation. The figure shows 5 kb of chromosome 13 surrounding the rhoptry-associated protein 1 (RAP1, PVP01_1338500). The PacBio reads (mapped to the negative strand and displayed in blue) support the presence of an unannotated intron (red box), leading to additional predicted coding sequences upstream of this intron, and a different protein than annotated in the genome (thick blue bars at the bottom). (**C**) Example of two isoforms with different predicted protein coding sequences. The top panel shows that blood-stage parasites express a transcript for cytochrome b5-like heme/steroid binding protein (PVP01_0716500) identical to the annotated protein-coding sequence (although with a shorter 5'-UTR). The middle panel shows that *P. vivax* sporozoites express this gene from a different start site (red box) resulting in a shorter transcript and a different predicted protein. (Note also the presence of an unannotated, and alternatively spliced, intron in the 3'UTR).

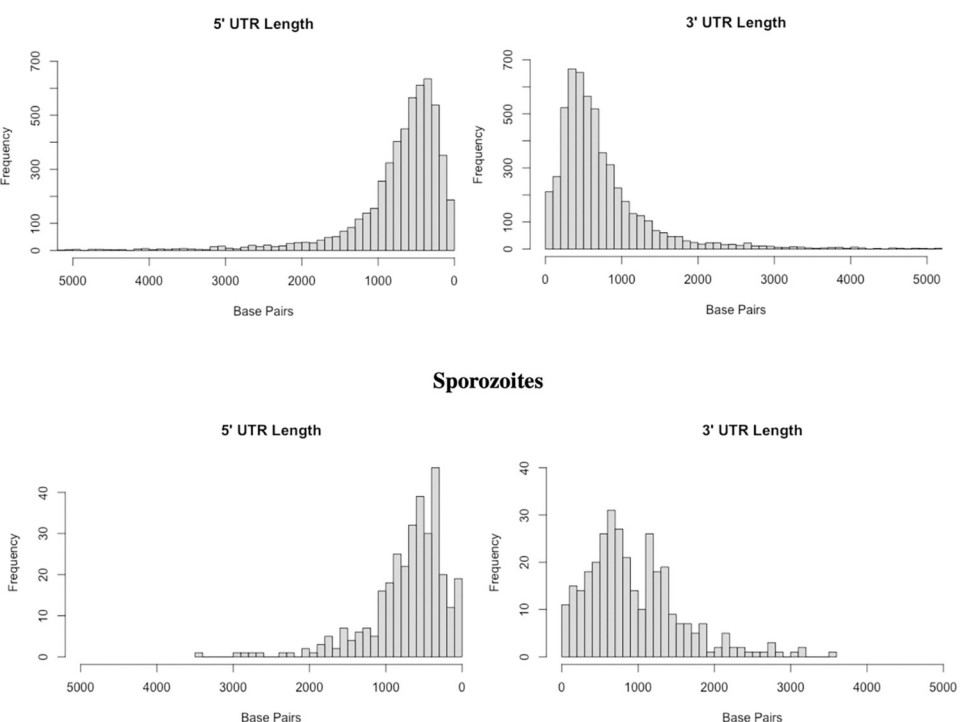

**Fig 3. UTR length distributions.** Distribution of UTR lengths (x-axis, in bp) for transcripts expressed by blood-stage parasites (top) and sporozoites (bottom).

respectively (**S1 Table**). Reanalysis on an independent scRNA-seq dataset generated from another *P. vivax* blood stage infection (AMRU-I from [24]) confirmed these results, improving the mapping of reads to annotated transcripts from 30% to 88%.

While UTRs have not been extensively studied in *Plasmodium*, in many eukaryotes they contain important regulatory elements which often affect mRNA stability [37–41]. While the length of the 5'-UTR does not appear to be associated with the level of the transcript expression determined using the short reads (p = 0.315), the 3'-UTR length was negatively correlated with expression level (p = $2.8 \times 10^{-6}$), although with a very low coefficient of correlation (Spearman's Rho = -0.0798) (**S8 Fig**). In an attempt to identify regulatory elements in the UTRs, we compared the abundance of all possible 5-mers in the 5'-UTRs, 3'-UTRs and promoter regions but failed to detect significant enrichment (**S9 Fig**). More in-depth analyses will be required to understand the function of these UTRs but the data presented here will provide a solid foundation to implement these studies.

Interestingly, out of the 5,368 full-length blood stage protein-coding transcripts, 1,072 (20%) had at least one intron in the UTRs: 419 transcripts had at least one intron in the 5'-UTR, 568 in the 3'-UTR and 89 had introns in both UTRs (see e.g., **S10 Fig**). Of these, 322 transcripts had more than one intron in their UTRs. Of the 344 full-length sporozoite transcripts, 26 had introns in the UTRs, 12 had at least one intron in the 5'-UTR, 11 in the 3'-UTR and 3 had introns in both UTRs. Five of these transcripts had more than one intron in the 5'- or 3'-UTR. Remarkably, in blood-stage parasites the levels of expression of the transcripts with an intron in the UTR were, on average, 3-fold higher than those of genes with no UTR intron (Wilcoxon rank test, p-value < $2.2 \times 10^{-16}$), suggesting that presence of introns in the UTR may be associated with increased mRNA stability in *P. vivax*. Furthermore, genes with introns in

their coding sequences were also three times more likely to have introns in the UTR ($\chi^2$ = 206.78, p-value < 2.2e-16), which may suggest that UTR splicing is mechanistically associated with splicing of coding sequences, possibly due to a more effective recruitment of the splicing machinery at those transcripts. To evaluate whether the presence of UTR introns was more frequent, at a specific developmental stage, we assigned each gene to the blood stage where it was most abundantly expressed. The proportion of genes with introns in their UTR was significantly different among stages ($\chi^2$ = 26.141, p-value = 8.9 x 10$^{-6}$), with 11% and 18% of the genes most expressed in early and late trophozoites having an UTR intron, respectively, compared to only 3% and 8% of the genes expressed in female gametocytes and schizonts (note that genes with multiple isoforms were excluded from this analysis as it is difficult to robustly quantify the relative expression of isoforms). Finally, we tested whether genes containing introns in their UTRs disproportionally belonged to a specific pathway but failed to detect any gene ontology enrichment (FDR < = 0.1, **S3 Table**).

## Transcript isoforms are common in P. vivax and can be expressed in a stage-specific manner

The 5,368 full-length protein coding transcripts derived from blood-stage parasites were transcribed from 2,869 genes: 1,687 genes (59%) were transcribed into a single isoform, while 1,182 (41%) showed evidence of multiple isoforms (and out of those, 719 genes were expressed in two isoforms, 247 in three and 216 were transcribed in four or more isoforms). Most isoforms (n = 819, 70%) encoded the same protein sequence and differed only in their UTR length: 1,024 (87%) genes with isoforms differed in their 5'-UTR length and 1,008 (85%) differed in the 3'-UTR length, with 450 genes with isoforms differing in their UTR introns (**Table 2**, see also **S10 Fig** for an example). 363 genes showed evidence of isoforms that were predicted to encode for different proteins, due to an alternate protein coding start (299), alternative end (311), and/or exon skipping (67) (**Table 2**).

In sporozoites, the 344 full-length protein coding transcripts were transcribed from 238 genes: 211 genes (89%) were transcribed into a single isoform, while 27 (11%) showed evidence of multiple isoforms (20 genes were expressed in two isoforms and three in three isoforms and four in four or more). All but eight isoforms were predicted to encode the same protein. Five of these genes had an alternate coding start and six had an alternate coding stop resulting in the change in coding sequence, and all had an exon skip or truncation (**Table 2**).

Since the full-length transcript data derived from molecules characterized by scRNA-seq, these data provide a unique opportunity to preliminarily examine whether different isoforms were expressed at different stages of the parasite development. Using the 10X cell barcodes, we determined the developmental age of each cell using pseudotime (determined from the Illumina scRNA-seq data). We only considered in this analysis 636 genes that had two (or more) isoforms expressed in at least 50 individual cells each. 123 (19%) of these genes showed evidence of expressing isoforms according to the parasite development (**S4 Table**). These stage-specific isoforms included 62 and 35 genes with differences in 5'- or 3'-UTR length (see e.g., **Figs 4** or **S11A**) and/or changes in coding sequences (n = 40) (**S11B Fig**). These findings suggest that *P. vivax* utilizes alternative start or termination of transcription as a means of transcriptional or translational regulation between stages.

**Table 2. Summary of isoform types from PacBio predictions.**

| Sample | No. of genes | Genes with >1 isoforms | Alt 5'-UTR | Alt 3'-UTR | Alt Coding Seq | Alt Coding Start | Alt Coding End |
|---|---|---|---|---|---|---|---|
| Blood | 2,870 | 1,182 | 1,024 | 1,008 | 363 | 299 | 311 |
| Sporozoites | 238 | 27 | 8 | 9 | 8 | 5 | 6 |

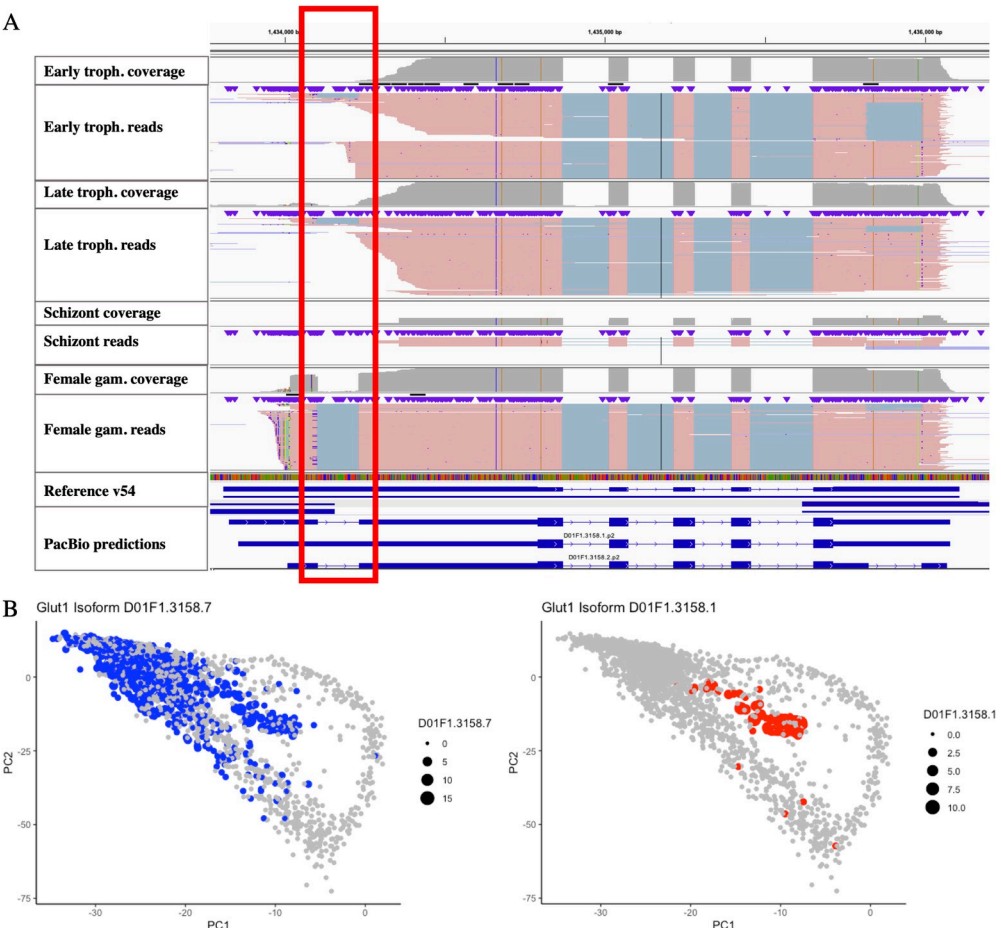

**Fig 4. Example of isoforms expressed in a stage-specific manner. (A)** Each panel shows the PacBio reads mapped to the glutaredoxin 1 (PVP01_0833900) and split in four groups according to the stage of the parasites they derived from: early trophozoites, late trophozoites, schizonts and female gametocytes (the terminology used for each group reflects developmental categories based on pseudotime analysis and might not exactly correspond to stages determined by microscopy). Female gametocytes express glutaredoxin 1 from a more upstream TSS than asexual parasites, and the resulting transcripts have an additional intron in the 5'-UTR (red box). (Note also the presence on an alternatively spliced intron in the 3'UTR of some transcripts). **(B)** PCAs showing that the short (blue) and long (red) isoforms for glutaredoxin-1 are expressed at stages of the parasite development, with the long isoform almost exclusively expressed in female gametocytes.

## Conclusion

Long read PacBio sequencing coupled with short read Illumina sequencing of single cell RNA-seq libraries provide a robust method to detect and characterize full-length transcripts and to identify mRNA isoforms expressed at different times. These simple modifications of existing laboratory protocols for generating 10XGenomics scRNA-seq can be easily applied to any eukaryotic cells and could be invaluable to examine variations in gene expression in organisms that are difficult to study in the laboratory and display a complex life cycle. This study also yielded novel insights on the variations and regulation of gene expression in *P. vivax*, which provides a solid framework to improve our understanding of *P. vivax* biology and regulation. This study notably highlighted the presence of extensive and incompletely annotated UTRs, and of ubiquitous UTR introns, that were sometimes expressed in a stage-specific manner. In particular, the observation that different stages used different start sites for transcribing the

same gene provides new insights on how gene expression may be regulated throughout the *Plasmodium* life cycle. In addition, the extensive variations observed in 3'-UTR lengths and the presence of introns in these regions might indicate important roles in mRNA stability or translation efficiency and, consequently, in the regulation of *Plasmodium* parasite biology.

## Materials and methods

### Ethics statement

All animal procedures were conducted in accordance with the National Institutes of Health (NIH) guidelines and regulations [42], under approved protocols by the National Institute of Allergy and Infectious Diseases (NIAID) Animal Care and Use Committee (ACUC) (Animal study NIAID LMVR15). Animals were purchased from NIH-approved sources and transported and housed according to Guide for the Care and Use of Laboratory Animals [42].

### Animal studies and sample collection

We infected two splenectomized *Saimiri boliviensis* monkeys with the Chesson strain of *P. vivax* using parasitized erythrocytes from cryopreserved stocks. Once they developed a parasitemia >0.1%, we collected 1 mL of blood from the femoral vein of each monkey after anesthesia with 10 mg/kg of ketamine and processed the blood samples on MACS LS columns as previously described [24]. Note that this enrichment procedure, that relies on the paramagnetic properties of hemozoin generated by maturing blood-stage parasites, fails to capture ring stage parasites.

Two blood samples from two additional *Saimiri boliviensis* monkeys coinfected with a NIH-1993 clone [24] and the Chesson *P. vivax* strain were used for membrane feeding of *Anopheles stephensi* and *Anopheles freeborni*. Salivary glands sporozoites were collected from each feeding at 21 days post-feed: 50 female mosquitoes were anesthetized on ice and their salivary glands dissected in PBS under a stereomicroscope. The salivary glands were transferred to a low-retention tube (Protein LoBind Tube; Eppendorf) containing PBS, homogenized with a disposable pestle, spun down, washed, resuspended, and quantified.

### 10X single cell RNA-sequencing library preparation and sequencing

An estimated 3,000 infected red blood cells or sporozoites from each sample were loaded onto 10X Chromium controller to prepare scRNA-seq libraries according to the manufacturer's instructions. We then generated, from each library, 57–75 million paired-end reads using an Illumina NovaSeq.

In addition, an aliquot of the cDNA prior to fragmentation was amplified by eight additional cycles of PCR before preparation of a PacBio library using the SMRTBell Express kit 2.0. We then generated 196–328 million reads from each library using a PacBio Sequel II.

### Short read analysis and single cell characterization

Following Illumina sequencing, the short reads were processed as described previously [24]. Briefly, we mapped all reads to the P01 *P. vivax* genome [36] using Hisat2 [43] with the default parameters except for a maximum intron length of 5,000 bp. We then removed PCR duplicates by identifying reads with identical barcode, unique transcript identifier (UMI) and mapping coordinates. We assigned each unique read to i) a cell based on its barcode and ii) a 500 bp window based on its genomic position. Only cells defined by more than 5,000 unique reads in blood stages and 250 unique reads in sporozoites were further analyzed. Count matrixes and principal component analysis was performed using in-house scripts.

## Long read analysis

The entire analytical pipeline for processing and analyzing the long-read data is described in S5 Fig. Briefly, the raw PacBio reads were first collapsed into circular consensus sequencing (CCS) using smrtanalysis [44] and only CCS supported by more than 10 passes were considered for further analysis. We then compared the 10X barcodes of each CCS reads with those obtained after Illumina sequencing and kept all reads matching the barcodes of one of the cells characterized by more than 5,000 unique Illumina reads. To account for the higher error rate of PacBio sequencing we allowed up to one nucleotide mismatch in the barcode sequence. After trimming the 10X adapters, 10X barcodes and the polyA tails using custom scripts, we mapped all CCS reads to the *P. vivax* P01 genome using minimap2 [45] using the cdna parameters and a *k* of 14. PCR duplicates were removed as described above. Only reads mapped to the P01 genome over 50% of their length were kept for further analysis.

## Transcript and protein identification

Transcript prediction was performed separately for each sample utilizing Stringtie2 [46,47]. Mapping files were divided by forward and reverse reads, and into single and multiple exon reads. Each of the four files was run separately using Stringtie2 long read default parameters. Only transcripts supported by more than 10 reads in each sample were considered for further processing (separately for the blood-stage and sporozoite samples). Transcript predictions were then compared and collapsed across samples into a single gtf for the blood-stage and sporozoite samples using gffcompare and custom scripts. We then used Transdecoder [48] to identify putative protein coding sequences from the predicted transcript sequences. Finally, we compared the predicted protein sequences to those annotated in the most current *P. vivax* P01 genome (v54) using BlastP and custom scripts.

## Stage-specific transcript analysis

To identify isoforms expressed in a stage-specific manner, we first assigned each PacBio read to a specific isoform with BLAT, using all predicted stringtie transcripts (including non-coding and incomplete transcripts) and considering the match with the greatest overall identity (and only considering reads aligned with >90% identity as aligned). We then identified the pseudo-time of the cell expressing this transcript by matching the GEMS barcode to the Illumina data. For each gene that had two (or more) isoforms, each expressed in more than 50 cells, we then tested for qualitative differences in isoform expression associated with the parasite development by comparing the pseudotime ranks of the cells expressing the first isoform with the pseudotime ranks of the cells expressing the second isoform using a Kolmogorov-Smirnov test.

## Supporting information

**S1 Fig. Example of Illumina and Pacbio data generated from the same scRNA-seq library.** The top panel shows the data generated by Illumina sequencing and displays typical peaks corresponding to the 3'-end of each expressed transcript. The middle panel shows data generated, from the same mRNAs, using PacBio sequencing and illustrates how generating full-length transcripts improves interpretation of scRNA-seq data for organisms with incomplete gene annotations.
(TIF)

**S2 Fig. Principal component analysis showing the relationships among individual parasite cells characterized by scRNA-seq (using the Illumina data).** Top row: each dot is a single cell

blood stage transcriptome, and is displayed based on its gene expression profile and colored according to the expression of stage markers (with cells being assigned to marker with the highest expression): red–early trophozoites, green–late trophozoites, purple–schizonts, turquoise–female gametocytes. Note that few ring stage parasites are included (if any) due to the enrichment method used (see *Material and Methods*). Bottom row: single cell sporozoite data.
(TIF)

**S3 Fig. Correlation between the number of reads obtained by Illumina and PacBio.** The scatterplot shows the correlation between the number of Illumina reads (x-axis) and PacBio reads (y-axis) obtained from each cell (individual black dots). Each panel represents the data for a different sample.
(TIF)

**S4 Fig. Predicted protein length distributions.** Distribution of the length (in amino acids) of the protein-coding sequences predicted from the PacBio transcripts (in red) and of the protein-coding sequences annotated in the P01 *P. vivax* genome (in blue).
(TIF)

**S5 Fig. Summary of the bioinformatic pipeline used for processing the PacBio reads and for predicting transcripts.**
(TIF)

**S6 Fig. Comparison of the predicted proteins with annotated proteins.** Distribution of the percentage alignment (x-axis) of the predicted protein coding sequences with the most similar protein sequence annotated in the P01 genome. Left: blood-stage transcripts. Right: sporozoite transcripts. (Note that the y-axis is cut and the right-most bars (perfect matches) go to 400 and 250 for the left and right panels, respectively).
(TIF)

**S7 Fig. Stage-specific expression of the isoforms of the cytochrome b5-like heme/steroid binding protein (PVP01_0716500).** The left PCA shows blood-stage and sporozoite parasites jointly displayed according to their gene expression profiles. The right figure shows the same PCA, with each parasite colored based on the cytochrome b5-like heme/steroid binding protein isoform expression: red–"sporozoite" isoform, blue–"blood-stage" isoform.
(TIF)

**S8 Fig. UTR length and expression.** Correlation between the length of a transcript's UTR (x-axis, in bp) and its level of expression determined by Illumina data (y-axis). Note that only genes expressing a single isoform are included in this analysis.
(TIF)

**S9 Fig. UTR kmer analysis.** Comparison of the abundance of all 5-mers in gene promoters, 5'-UTRs and 3'-UTRs. Note that most motifs with different abundance (i.e., deviating from the diagonales) are either repeated sequences or encoding for a start codon (ATG).
(TIF)

**S10 Fig. Example of a transcript with unannotated UTR introns.** The figure shows PacBio reads (in blue) corresponding to the annotated mRNA for coenzyme Q-binding protein COQ10 homolog (PVP01_0113000) but with an unannotated intron in the 3'-UTR (red box) as well as, for a subset of the mRNAs, a second intron in the 5'-UTR (blue box).
(TIF)

**S11 Fig. Examples of isoforms expressed in a stage-specific manner.** Each panel shows the PacBio reads mapped to a selected locus and split in four groups according to the stage of the parasites they derived from: early trophozoites, late trophozoites, schizonts and female gametocytes (from top to bottom). **(A)** Early trophozoites express the Ham 1-like protein (PVP01_0316500) from a more upstream TSS than the other stages and the resulting transcripts have a longer 5'-UTR containing five introns (red box). **(B)** The isoforms expressed from suppressor of kinetochore protein 1 (PVP01_1105000) result into different predicted protein coding sequences: some transcripts expressed exclusively in early trophozoites retain the third intron (red box) leading to a different open reading frame (blue bars at the bottom). (TIF)

**S1 Table. Sequencing and transcript predictions results.** Table listing numbers from each step of sequencing analysis and transcript prediction. (XLSX)

**S2 Table. List of all predicted transcripts.** Table listing all predicted transcripts, their nucleotide sequence, amino acid sequence (if applicable), most similar PVP01 gene name and description (if applicable), their UTR lengths, and the number of introns. (XLSX)

**S3 Table. Results of Gene Ontology analysis.** Top hits from GO analysis. (XLSX)

**S4 Table. List of predicted genes with differentially expressed isoforms.** (XLSX)

**S1 Data. Gene predictions.** gtf file of all predicted protein coding transcripts combinned with the current reference. (GTF)

## Acknowledgments

We thank the technicians, care-takers, and veterinaries of the Division of Veterinary Resources and of the insectary of the Laboratory of Malaria and Vector Research, National Institute of Allergy and Infectious Diseases, for animal care, technical assistance, and mosquito rearing; and S. Ott, H. Bowen, L. Sadzewicz and L. Tallon in the Genomic Resource Center at the University of Maryland School of Medicine for their support with Illumina and PacBio sequencing.

## Author Contributions

**Conceptualization:** Thomas E. Wellems, David Serre.

**Data curation:** Brittany Hazzard.

**Formal analysis:** Brittany Hazzard.

**Funding acquisition:** Thomas E. Wellems, David Serre.

**Investigation:** Brittany Hazzard, Juliana M. Sá, Angela C. Ellis, Tales V. Pascini, Shuchi Amin.

**Methodology:** Juliana M. Sá, Tales V. Pascini, David Serre.

**Project administration:** Thomas E. Wellems, David Serre.

**Resources:** Juliana M. Sá, Tales V. Pascini, Thomas E. Wellems.

**Software:** Brittany Hazzard.

**Supervision:** Thomas E. Wellems, David Serre.

**Validation:** Brittany Hazzard.

**Visualization:** Brittany Hazzard.

**Writing – original draft:** Brittany Hazzard, David Serre.

**Writing – review & editing:** Brittany Hazzard, David Serre.

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
