## [Decision Letter · Decision Letter 0]

23 Aug 2022

Dear Ms Hazzard,

Thank you very much for submitting your manuscript "Long read single cell RNA sequencing reveals the isoform diversity of Plasmodium vivax transcripts" for consideration at PLOS Neglected Tropical Diseases. As with all papers reviewed by the journal, your manuscript was reviewed by members of the editorial board and by several independent reviewers. In light of the reviews (below this email), we would like to invite the resubmission of a significantly-revised version that takes into account the reviewers' comments. 

We cannot make any decision about publication until we have seen the revised manuscript and your response to the reviewers' comments. Your revised manuscript is also likely to be sent to reviewers for further evaluation.

Sincerely,

Ananias A. Escalante, PhD

Academic Editor

Ana Rodriguez

Section Editor

Reviewer's Responses to Questions

**Key Review Criteria Required for Acceptance?**

**Methods**

-Are the objectives of the study clearly articulated with a clear testable hypothesis stated?

-Is the study design appropriate to address the stated objectives?

-Is the population clearly described and appropriate for the hypothesis being tested?

-Is the sample size sufficient to ensure adequate power to address the hypothesis being tested?

-Were correct statistical analysis used to support conclusions?

-Are there concerns about ethical or regulatory requirements being met?

Reviewer #1: The data collection methods are clear, although more background could be given to put the methods in context. The code used for the analysis is not available on the github link provided.

Reviewer #2: The methods are adequately described.

Reviewer #3: Minor revisions only ( see below)

**Results**

-Does the analysis presented match the analysis plan?

-Are the results clearly and completely presented?

-Are the figures (Tables, Images) of sufficient quality for clarity?

Reviewer #1: The results are clearly presented.

Reviewer #2: The results are clearly and concisely presented.

Reviewer #3: Minor revisions only ( see below)

**Conclusions**

-Are the conclusions supported by the data presented?

-Are the limitations of analysis clearly described?

-Do the authors discuss how these data can be helpful to advance our understanding of the topic under study?

-Is public health relevance addressed?

Reviewer #1: Conclusions could be strengthened if they can show that their data can be used to improve the reference and increase mapping rate.

Reviewer #2: The conclusion section is remarkably brief and could be extended. The authors could mention the applicability of this methodology to other parasites, the possible significance of alternative mRNA isoforms and how this could be followed up and how the findings of this study improve our understanding of P. vivax biology.

Reviewer #3: Minor revisions only ( see below)

**Editorial and Data Presentation Modifications?**

Reviewer #1: Minor Comments

-The authors should cite papers outside of malaria that have used this technology before, there is no context given for the development of the technology and what it has been used for in other systems.

- The authors justify using pacbio to look at isoforms because the 10x libraries are 3' biased. However, they don't show that this is actually the case for P. vivax. Could the authors show this coverage bias in their data?

-Fig S1 Blood stages: show expression of markers so it is clear how stage was assigned.

 Sporozoites: Colour the plot by a marker gene or UMI. Is there a subset of higher quality cells that could be used

-How have you controlled for different levels of expression in the stage-specific transcript analysis?

Reviewer #2: I have only minor comments for the authors to consider.

1. On line 147, the authors state that the length of the transcripts from asexual stages ranged from 100 to 20,506 bp. A 20 kb transcript is remarkably long, much longer than any annotated gene. What is this transcript and do the authors think this is a genuine mRNA?

2. In Figures 2C, 5A and 5C there appears to be evidence for alternative splicing that is not mentioned in the Figure legend. Is this correct? If so, it might be worth mentioning so that readers are not confused. 

3. In Figure 5, the authors mark the figures with the terms “Group A”, “Group B” and “Group C”, while the figure legend explains that these terms refer to early trophozoites, late trophozoites and schizonts respectively. Why not simply mark the figure itself with the stage of the parasites? I don’t see the utility of using the terms “Group A, B, C”.

Reviewer #3: (No Response)

**Summary and General Comments**

Reviewer #1: Hazzard et al have generated short- and long-read scRNAseq data using the standard 10x genomics pipeline in combination with pacbio sequencing in P. vivax blood stages and sporozoites. Their analysis identifies a subset of transcripts that do not match the current reference annotations as well as isoforms that show a stage specific pattern of expression. They identify different UTR lengths between blood-stage parasites and sporozoites, and UTRs that contain introns, which they hypothesize might play a role in gene regulation.

Overall, I found the analysis superficial and lacking of interesting biology. Although this is an interesting dataset, the authors have not invested enough in the analysis to either reveal noteworthy biology or provide the community with a valuable resource dataset. I suggest strengthing the paper as a reference dataset prior to publication.

Major Comments

The authors show that only half (or less) of the reads map within 500 bp of an annotated p. vivax gene and suggest that this is because of incomplete annotation of the reference genome. Can the authors use their data to improve the reference and demonstrate that more of the reads would map if they had better annotations?

line 196: "More in-depth analyses will be required to understand the function of these UTRs but the data presented here will provide a solid foundation to implement these studies." If the authors want this to be a useful dataset they need to provide the actual data. Could the authors provide a gtf with the newly annotated isoforms and UTRs? Can the dataset be integrated into PlasmoDB?

-Three of the five main figures are screen shots from a genome browser tool. I suggest that the authors diversify and include more information in the main figures. For example, where they demonstrate different isofroms between blood stage and sporozoite parasites, what does expression of the different isoforms look like on the PCA and does that help them interpret the biology of the cell clusters within the blood stages?

Reviewer #2: The submission by Hazzard and colleagues describes the application of combined Illumina/PacBio sequencing to 10X single cell transcriptomic analysis of Plasmodium vivax infected red blood cells. P. vivax remains a major cause of morbidity in large regions of the developing world, however analysis of these parasites has been hampered by the inability to raise this species in culture, thus limiting experimental approaches. The application of single cell transcriptomics to parasites isolated from patients or from experimentally infected primates has enabled a much better analysis of gene expression patterns, however technical limitations have resulted in most information being derived from sequences near the 3’ ends of the transcripts. Here the authors attempt to address this issue by using long-read sequencing technology, thereby deriving sequence information from much longer stretches of the transcripts, in many cases obtaining full-length transcript information. This enabled them to identify products of alternative splicing and variations in the 5’ and 3’ UTRs. In addition, by identifying the developmental stage of each individual cell analyzed, they were able to associate stage-specificity with alternative isoforms, something that was difficult previously. 

The manuscript reports an important and useful technical advance in the study of Plasmodium vivax. The methodology described here is likely to be applied by many research groups in the future, and the advance will likely also be applicable to other Plasmodium species, including P. falciparum. The manuscript also includes novel discoveries regarding alternative mRNA isoforms, which will undoubtedly be following up to determine their biological significance. Overall, the manuscript is well written and easily understood.

Reviewer #3: The manuscript by Hazzard and colleagues presents a single-cell sequencing dataset consisting of Plasmodium vivax blood stages and sporozoites, sequenced using two different platforms – traditional Illumina sequencing and PacBio long-read sequencing. 

Plasmodium vivax remains a relevant human pathogen and given how few good quality transcriptomics datasets are available, the presented work is certainly of significant value to the parasitology community. Additionally, the combination of the two techniques provides a wealth of information regarding the parasite transcript structure, gene expression changes etc. The rationale of the work is clearly stated and backed up by relevant literature. The manuscript is generally clearly written.

There is however a number of details (mostly related to data analysis) that should be clarified in order to justify some of the conclusions of the manuscript. The major ones are:

1) Cell identity: 

It is not clear how different stages of the parasite life cycle were identified in the blood sample. The single-cell data appears to be plotted as PCA components (rather than UMAP or t-SNE graphs which are the golden standard of dimensionality reduction for this data type) and does not form clear clusters. There is no mention of any attempts of clustering, identification of marker genes or developmental trajectory mapping etc. Pseudotime analysis is mentioned but there are no methods attached to it, so it is unknown what was done exactly. The sporozoite samples are processed separately so it is not clear how they relate to other stages. Given the importance for further transcript assignment, the details of this analysis (if done) should be included, ideally pooling the sporozoite samples together with other stages.

2) Full transcript annotation: 

Authors mention themselves that numerous transcripts/peptides predicted from their annotation appear to be truncated blaming it on “the 10X reverse transcriptase processivity that will hamper synthesis of long cDNA molecules and prevent recovery of full-length transcripts”. In this light, the fact that the vast majority of the new isoforms they identified differ in the length of 3’UTR or 5’UTR only, may suggest that they represent truncated RNA molecules rather than full functional transcript isoforms. The fact that UTRs in general have different lengths between the stages (something not observed in the other Plasmodium datasets as far as I know) may support this. 

In this case more stringent filtering conditions (or an attempt to validate at least one of the "new' transcripts by eg. Northern blot, although I realise that authors may not have sufficient material or the tools to attempt it) need to be used to identify the novel Plasmodium isoforms

This should be acknowledged in the manuscript and the general conclusions regarding the transcript UTRs length should be toned down.

3) Differential isoform usage between the life stages. 

Again, it is not clear how robust this analysis is without further details. Rather than using tools for isoform usage quantification, the authors choose to go for custom approach consisting of the stage distribution of cells containing a given isoform across the life cycle (if I understand correctly). No details of this method and no processed output is given. The manuscripts is badly missing a table with the list of all stage-specific isoforms, the numbers of molecules that their analysis is based on, and the results of statistical test applied to each of them. More details (or validation of one of the genes with >1 isoforms) are required to substantiate the authors claim regarding the stage-specificity.

Additionally:

1) there are no attempts to compare the author’s dataset to similar ones generated for P.falciparum, a model human malaria species. As the two parasites share very similar syntenic genomes (with 1:1 orthologs for almost all genes) and similar exon/intron structure, such comparison could be very useful, given the fact that P.f has much better UTR annotation and a number of stage specific gene expression studies 

2) The authors don’t provide the link to raw datasets (eg. SRA, GEO).Given the declaration in the data availability statement, I assume that this is a last-minute omission and the data will be made accessible to the community

3) A Github repository with the scripts for the data analysis is empty. 

4) it is not clear what the fig S7 should represent as the labelling of the axes is not very helpful. No statistical test is applied, and no raw numbers are given, so it is difficult to tell how significant the difference in kmers is. For the motif enrichment in the specific sequences, a number of standard tools are available (Meme, Ace etc.)

4. In asexual stage annotation authors identify the trophozoites, schizonts and females but not the rings and males which should be also present in the mixed blood stages. Was that an artefact of the experimental design? Can authors provide reasons for that?

5. Based in the few examples shown in Fig 5, the transcripts with multiple isoforms tend to be more expressed at the stage in which a new isoform is identified. Given how many genes are differentially expressed between the life stages, it would be useful to include the results of differential expression analysis of the genes which are differentially spliced in order to see if one phenomenon I connected with the other (eg. is the increase of expression correlated with different UTR). That could also provide a hint whether some of the isoforms are just result of faulty transcript processing that appear whether the gene is highly expressed.

Regardless of these omissions, the study is well designed and its publication will be of interest for the malaria community. I commend the authors for their effort and looking forward to seeing this work in print.

PLOS authors have the option to publish the peer review history of their article (what does this mean?). If published, this will include your full peer review and any attached files.

Reviewer #1: No

Reviewer #2: No

Reviewer #3: No
---

## [Decision Letter · Decision Letter 1]

28 Nov 2022

Dear Ms Hazzard,

We are pleased to inform you that your manuscript 'Long read single cell RNA sequencing reveals the isoform diversity of Plasmodium vivax transcripts' has been provisionally accepted for publication in PLOS Neglected Tropical Diseases.

Best regards,

Ananias A. Escalante, PhD

Academic Editor

Ana Rodriguez

Section Editor

Reviewer's Responses to Questions

**Key Review Criteria Required for Acceptance?**

**Methods**

-Are the objectives of the study clearly articulated with a clear testable hypothesis stated?

-Is the study design appropriate to address the stated objectives?

-Is the population clearly described and appropriate for the hypothesis being tested?

-Is the sample size sufficient to ensure adequate power to address the hypothesis being tested?

-Were correct statistical analysis used to support conclusions?

-Are there concerns about ethical or regulatory requirements being met?

Reviewer #1: (No Response)

Reviewer #3: No further comments, all my previous ones were adressed.

**Results**

-Does the analysis presented match the analysis plan?

-Are the results clearly and completely presented?

-Are the figures (Tables, Images) of sufficient quality for clarity?

Reviewer #1: (No Response)

Reviewer #3: Yes, perhaps more details could be present in the legend of Table S4 ( what is D-crit test?)

**Conclusions**

-Are the conclusions supported by the data presented?

-Are the limitations of analysis clearly described?

-Do the authors discuss how these data can be helpful to advance our understanding of the topic under study?

-Is public health relevance addressed?

Reviewer #1: (No Response)

Reviewer #3: Yes

**Editorial and Data Presentation Modifications?**

Reviewer #1: (No Response)

Reviewer #3: No further problems

**Summary and General Comments**

Reviewer #1: The authors have adequately addressed my previous comments. This will be a useful resource for the malaria community.

Reviewer #3: No additional experiments or analysis needed

PLOS authors have the option to publish the peer review history of their article (what does this mean?). If published, this will include your full peer review and any attached files.

Reviewer #1: No

Reviewer #3: No

---

## [Editor Report · Acceptance letter]

12 Dec 2022

Dear Ms Hazzard,

We are delighted to inform you that your manuscript, "Long read single cell RNA sequencing reveals the isoform diversity of Plasmodium vivax transcripts," has been formally accepted for publication in PLOS Neglected Tropical Diseases.

Best regards,

Shaden Kamhawi

co-Editor-in-Chief

Paul Brindley

co-Editor-in-Chief
